# The STRENGTH Study: A cluster randomised controlled trial of the effect of a behaviour change intervention added to cardiac rehabilitation on physical activity adherence

Clare T. M. Doherty[1], Mark A. Tully[2], Jason J. Wilson[3], Leonie Heron[4], Helen McAneney[2], Victoria Irving[5], Lisa Spratt[6], Rachel O'Reilly[7], Kim Kensitt[8], Nicole E. Blackburn[1]*

1 School of Health Sciences, Ulster University, Derry~Londonderry, United Kingdom, 2 School of Medicine, Ulster University, Derry~Londonderry, United Kingdom, 3 School of Sport and Exercise Science, Ulster University, Derry~Londonderry, United Kingdom, 4 Institute of Social and Preventative Medicine, University of Bern, Bern, Switzerland, 5 Better Belfast, Greenwich Leisure Limited, Belfast, United Kingdom, 6 National Health Service, Belfast Health and Social Care Trust, Belfast, United Kingdom, 7 South Eastern Health and Social Care Trust, Lisburn, United Kingdom, 8 Belfast Health Development Unit, Public Health Agency, Belfast, United Kingdom

* ne.blackburn@ulster.ac.uk

## Abstract

### Background

Coronary heart disease is the leading cause of global mortality, imposing significant health and economic burdens. Cardiac rehabilitation, including physical activity, can reduce coronary heart disease-related morbidity and mortality. We tested whether the addition of a behaviour change intervention to cardiac rehabilitation could promote and maintain physical activity achieved during cardiac rehabilitation, beyond standard care timeframes.

### Methods

A cluster randomised controlled trial was conducted across six community-based maintenance stage cardiac rehabilitation classes. A total of 96 participants (mean age 65.04 ± 8.38 years; 75% male) received either standard care or a behaviour change intervention, with physical activity, measured with an ActiGraph GT3X+ accelerometer as the primary outcome.

### Results

No significant differences in daily minutes of moderate-to-vigorous physical activity and steps per day, or any secondary outcomes, including self-rated health, quality of life, and mental wellbeing, were observed between the intervention and control groups at six months follow-up. These findings suggests that the behaviour change

**Data availability statement:** All relevant data are within the paper and its Supporting information files.

**Funding:** This study was supported by Heart Research United Kingdom: RG2686. The funders had no role in study design, data collection and analysis, decision to publish, or preparation of the manuscript.

**Competing interests:** The authors have declared that no competing interests exist.

**Abbreviations:** Δ, Change in; ANOVA, Analysis of Variance; BACPR, British Association for Cardiovascular Prevention and Rehabilitation; BHF, British Heart Foundation; BMI, Body Mass Index; BP, Blood Pressure; CHD, Coronary Heart Disease; CR, Cardiac Rehabilitation; DoH, Department of Health; EQ-5D-3L, EuroQol - 5 Dimensions - 3 Levels; EQ-VAS, EuroQol - Visual Analogue Scale; HR, Heart Rate; ICC, Intra-class Correlation Co-efficient; IQR, Interquartile Range; MCS, Mental Component Score; MSEQ, Marcus Self-Efficacy Questionnaire; MVPA, Moderate-Vigorous Physical Activity; NA, Not Applicable; PA, Physical Activity; PCS, Physical Component Score; SD, Standard Deviation; SF-12v2, Short-Form 12 version 2; STRENGTH, Self-management and Theory-based Rehabilitation Encouraging New Gateways to Healthy-Hearts; SW-RCT, Stepped-Wedge Randomised Controlled Trial; WEMWBS, Warwick Edinburgh Mental Wellbeing Scale; WHO, World Health Organization.

intervention did not significantly impact physical activity or health outcomes during maintenance cardiac rehabilitation. This may be attributed to high baseline physical activity levels among participants, and the extended cardiac rehabilitation support provided to both groups, potentially masking any intervention effects.

## Conclusion

A behaviour change intervention added to standard maintenance stage cardiac rehabilitation did not improve physical activity or health outcomes. However, continued access to cardiac rehabilitation sustained high physical activity levels. Future research should disentangle the independent effects of behaviour interventions and ongoing cardiac rehabilitation support.

## Trial registration

ClinicalTrials.gov NCT05705310

## Background

Cardiovascular diseases, including coronary heart disease (CHD), remain the leading cause of global morbidity and mortality, posing a major public health and economic challenge. Globally, an estimated 626 million people were living with CVD in 2023, resulting in 19.2 million deaths, a substantial rise from 1990, underscoring the growing global health burden of CVD [1]. CVDs accounted for 437 million disability-adjusted life years (DALYs) in 2023, with ischemic heart disease being the leading contributor. Most of this burden is attributable to modifiable risk factors, including high blood pressure, unhealthy diet, elevated cholesterol, physical inactivity, and other lifestyle-related risks, which together account for nearly 80% of CVD DALYs [1]. The rising prevalence of CVD, driven by population growth and aging, highlights the urgent need for interventions that support long-term management and lifestyle modification.

WHO (2025) reports that CVDs caused approximately 19.8 million deaths in 2022, accounting for 32% of all global deaths, with heart attack and stroke responsible for 85% of these [2]. Over three-quarters of CVD deaths occurred in low- and middle-income countries, and 38% of premature deaths (<70 years) from noncommunicable diseases were due to CVD. Most CVDs are preventable through modification of behavioural and environmental risk factors, including tobacco use, unhealthy diet, physical inactivity, harmful alcohol use, and air pollution. Early detection and management remain critical to reducing morbidity and mortality.

In the United Kingdom (UK), CHD is responsible for approximately 66,000 deaths annually, of which 25,000 are premature deaths of people <75 years old [3]. Annual UK health care costs are estimated at £9bn, including primary care, outpatient appointments, accident and emergency attendances, inpatient care, pharmacology, with other costs related to lost productivity from morbidity, mortality, and informal care [4]. Approximately 2.3m people are living with CHD in the UK [3]. The burden of CHD is set to rise [5], and as a result, interventions to reduce the burden of CHD on

society are needed [6]). Guidelines recommend that all patients should be offered cardiac rehabilitation (CR) following a cardiac-related event, surgery, or procedure [7].

The core CR programme (previously known as Phase III CR), offered to CHD patients following their discharge from hospital, includes lifestyle recommendations such as advice and support on daily alcohol consumption; achieving and maintaining a healthy weight; healthy diet; smoking cessation; and physical activity (PA), in line with national guidelines which recommend at least 150 minutes of moderate-intensity aerobic activity, 75 minutes of vigorous-intensity aerobic activity, or a combination of both, in addition to twice weekly strength training exercise [8,9]. CR appears to reduce early mortality risk, morbidity, and unplanned hospital admissions, along with improving quality of life and psychological well-being [10]. PA, which is an important component of CR, has been shown to reduce hypertension, raise levels of high-density lipoprotein cholesterol and lower low-density lipoprotein cholesterol [11]. Furthermore, PA assists in the control of HbA1c; a marker of long-term glycemic control [12]. In the UK, almost 10% of CHD deaths are attributable to physical inactivity [13]. In a review comparing exercise with drug interventions for the secondary prevention of CHD, findings demonstrated that the effects of exercise on mortality did not differ from commonly prescribed medications such as statins, β-blockers and angiotensin-converting enzyme inhibitors [14]. After the core CR programme has been completed, it is recommended that patients are invited to attend community-based exercise programmes, known as the maintenance stage (previously known as Phase IV CR).

On completion of the formal, structured CR, individuals in the UK are advised of relevant exercise programmes within their local leisure and community centres. While participation in CR has been found to improve PA levels [15,16], a number of studies suggest that changes in PA following completion of the CR programme are unlikely to be sustained independently beyond the formal CR period [17,18]. CR aims not only to improve physical fitness through structured exercise but also to increase habitual PA levels through education, self-management support, and behaviour change strategies. While supervised exercise in core CR programmes effectively improves fitness and short-term PA, sustaining these gains independently after the formal programme is challenging. Barriers such as lack of motivation, low self-efficacy, environmental constraints, and competing life demands often limit long-term adherence to PA recommendations [17–19]. Consequently, maintenance of PA following CR requires interventions that explicitly incorporate behaviour change techniques, such as goal setting, action planning, self-monitoring, and problem-solving, to support habit formation and encourage autonomous motivation. These strategies form the theoretical basis for the STRENGTH (Self-management and Theory-based Rehabilitation Encouraging New Gateways to Healthy-Hearts) study, which seeks to enhance sustained PA in patients completing maintenance stage CR.

Interventions that incorporate behaviour change strategies, including self-efficacy enhancement, problem-solving skills, and relapse prevention strategies, could establish maintenance of positive behaviour changes following maintenance stage CR [19]. Therefore, the STRENGTH study was developed. The aim of this study was to evaluate the effectiveness of a multi-component intervention incorporating current evidence from previous research and applying it to a maintenance stage CR programme to promote sustained changes in PA following completion. The hypothesis is that, following a CR programme, any PA changes would be more likely to be maintained by those who received the additional behaviour change programme promoting self-efficacy, compared to those who have received standard care.

## Methods

### Study design

The study was originally planned as a stepped-wedge randomised controlled trial (SW-RCT); however, due to operational constraints, including post-pandemic service pressures, the planned stepped-wedge rollout could not be fully implemented. As a result, the design aligns most closely with a cluster randomised trial with a stepped implementation sequence, rather than a full stepped-wedge trial. Randomisation occurred at the level of the CR class during one step of the rollout, and all participants received the condition allocated to their class.

The methods are reported in accordance with the CONSORT checklist [20] for SW-RCT (S1 File), to ensure comprehensive reporting, while acknowledging that a formal stepped-wedge analytic structure was not applied. This design was employed to test the implementation and effect of the addition of a self-management behaviour change programme to current maintenance stage CR. Stepped-wedge designs are parallel cluster trial designs, which are commonly used for the evaluation of service delivery and minimise the required sample size [21–23].

## Recruitment

All participants taking part in maintenance stage CR programmes, from six CR classes across two counties in N. Ireland, from 7 March 2022 to 15 February 2023, were invited to take part in the study. Recruitment occurred later than originally planned due to operational constraints, including post-pandemic service pressures and local service restructuring, which affected the timing of class availability and participant enrolment. Prior to this maintenance stage CR programme, they had completed the core CR and had been assessed and risk stratified by clinical staff following the British Association for Cardiovascular Prevention and Rehabilitation (BACPR) guidelines (2023) [7]. Due to COVID-19–related service pauses and delays, the interval between completion of core CR and enrolment in the study varied considerably, ranging from approximately 3 months to up to 2 years post–cardiac event. A postdoctoral researcher attended a session during the initial weeks of the maintenance stage CR programme to provide information about the study. In addition, information sheets were handed out by the CR nurses at the end of the core programme. Potential participants had the opportunity to ask any questions about the study and were assured of confidentiality. Those who agreed to participate were asked to sign a consent form. As part of the referral process to a maintenance stage CR programme, individuals were assessed for the suitability to participate in regular exercise. Therefore, only individuals who did not wish to provide consent to participate in the study or could not commit to the full 12 weeks of the CR programme, were excluded.

## Allocation and randomisation

Upon enrolment (which occurred typically in the first weeks of the programme and no later than week 3), participants' baseline measures were collected. Baseline data collection corresponded with the time initial monitoring was normally completed within maintenance stage CR programmes. During the first recruitment period of the study (Months 0–3), each of the four initial participating CR classes were allocated to the control group (Table 1). In the second recruitment period (Months 3–6), three of five participating CR classes were randomised to receive the intervention. In the final recruitment period (Month 6 onwards), all six participating CR classes received the intervention. Randomisation was conducted at the level of the CR class and was relevant during the second recruitment period only. The randomisation sequence was generated by a separate researcher (MAT), not involved in the delivery of the intervention or collection of outcomes. Allocation

**Table 1. Allocation of condition at each recruitment period of the study, for each participating CR class.**

| CR Class | Recruitment Period 1 (Months 0–3) | Recruitment Period 2 (Months 3–6) | Recruitment Period 3 (Months 6+) |
|---|---|---|---|
| 1 | Control | Control | Intervention |
| 2 | Control | Intervention | Intervention |
| 3 | Control | Intervention | Intervention |
| 4 | Control | Control | Intervention |
| 5 | NA | Intervention | Intervention |
| 6 | NA | NA | Intervention |

Allocation of condition followed a stepped-wedge design and was randomised at the level of the CR class during the second recruitment period. NA denotes that the CR class was not operating, or receiving referrals, during that corresponding recruitment period.

Abbreviations: Cardiac Rehabilitation (CR), Not Applicable (NA).

to intervention or control followed a predefined staggered implementation schedule rather than participant or staff choice, minimising allocation bias while recognising the potential for residual selection bias. Condition was randomised at the level of the CR class during the second recruitment period. Not applicable (NA) denotes that the CR class was not part of the study during that corresponding recruitment period. All participants were informed of their allocation to either the intervention or control group (i.e., standard care) after baseline assessment.

## Control

Individuals in the control condition received standard care (maintenance stage CR). The typical programme included two supervised classes per week, each lasting approximately 60 minutes, delivered in-person by leisure-centre exercise professionals trained in CR delivery. Sessions comprised a structured exercise component including warm-up, aerobic activity (e.g., treadmill, cycling, step-based routines), resistance exercises, and cool-down. Exercise intensity was guided by national CR recommendations (moderate intensity; Borg RPE 11–14), with individual tailoring based on functional capacity and clinical history. After the last outcome measures were collected at their 6-month follow-up, they were given a pedometer and shown how to use it.

## Intervention

Individuals in the intervention condition received a behaviour change programme (structure informed from a previous study [24]) alongside their maintenance stage CR (TIDieR checklist included in S2 File). The intervention was designed to enhance self-efficacy, motivation, action planning, and relapse prevention using established behaviour change techniques (BCTs), including goal-setting, self-monitoring of behaviour (step counts, activity logs), problem-solving and overcoming barriers, social support strategies, action planning, and positive reinforcement/feedback. During Week 6 of their maintenance stage CR programme, consenting participants were shown how to wear and use a validated pedometer (Yamax Digiwalker CW-701, Yamax Inc, Japan). Written instructions were provided in cases where in-person instruction was not possible. Participants were instructed to wear the pedometers during waking hours, except for during water-based activities. In accordance with advice given during their CR programme, participants were encouraged to aim to gradually build up to achieving the current national PA guidelines [8]. All participants continued to attend their standard maintenance CR sessions in local leisure centres alongside the behaviour change programme. Attendance at these sessions was recorded to monitor ongoing participation, although it was not included as a covariate in analyses due to highly variable attendance and minimal contribution of class-level clustering to outcome variance.

Participants were asked to self-monitor their activity by wearing the pedometer and recording their daily step counts and/or time in PA each week in activity diaries, which were also provided. If the participant already tracked their PA using a personal wearable device and expressed a preference to track their activity with their own device instead, they were encouraged to do so. Participants would review their progress with the researcher at the CR session (or through a phone call where in-person review was not possible) each week for six weeks. During reviews, the researcher helped study participants to set realistic step-count goals for the following week (e.g., 10% increase on previous weekly average). After the first six weekly reviews, which were designed to coincide with the end of the typical twelve-week CR programme, participants received three further monthly check-ins with the researcher, to check progress and encourage them to continue with their activities.

During the intervention, group discussions took place with the researcher and the participants. The researcher led the discussions, focusing on the benefits of regular PA, demonstrated lifestyle activities that could help accumulate activity and identified various means of social support. The group discussions were also used to identify barriers and facilitators of PA, and the groups were encouraged to troubleshoot any identified barriers together. Local opportunities (groups or places) for PA were identified, and participants were encouraged to try new types of PA. The intervention was designed to strengthen self-efficacy and self-regulation as the primary mechanisms to influence PA behaviour. Following the measurement of outcomes at six months after baseline, participants received no further contact from the researcher.

## Outcome measures

**Primary outcomes.** All participants outcomes were measured at baseline, twelve weeks (end of maintenance stage CR programme) and six months after baseline. The primary outcomes were: 1) steps per day, and 2) time spent in moderate-vigorous PA (MVPA) per day. These primary outcomes were measured using a validated ActiGraph wGT3X-BT accelerometer (ActiGraph LLC, Pensacola, FL, USA), which has been frequently used in patients with cardiovascular disease [25]. Participants were asked to wear the accelerometer placed on their dominant side on an elasticated belt, during waking hours only, for seven consecutive days at each of the time points [26], and were given written instructions and an activity monitor wear log to fill in. Accelerometer data were collected using the default setting of 30 Hz, and without the low-frequency extension filter being applied. Accelerometer data were processed using ActLife v6.13.4, with the criteria developed by Choi and colleagues [27] used for wear-time validation, with data summarised in 10-second epochs and scored using Troiano Adult criteria for MVPA using the single axis [28]. To be considered a valid day which was reflective of each participant's normal activity, the following inclusion criteria were applied in our analyses: minimum wear-time of 10 hours/day, on at least four days to include at least one weekend day.

**Secondary outcomes.** Changes in self-efficacy were measured using the PA self-efficacy scale [29] to evaluate the extent to which the intervention followed the anticipated behavioural process. Physical and mental health were measured using the short-form 12 version 2 (SF-12v2) Health Survey [30], and mental wellbeing was further assessed using the Warwick-Edinburgh Mental Well-Being Scale [31]. Health-related quality of life was measured using EQ-5D-3L [32], which included EuroQol Visual Analogue Scale (EQ-VAS), a self-rated measure of perceived health. Demographic information was collected at baseline via questionnaire, and physical measures consisting of blood pressure (OMRON M3 Comfort, OMRON Healthcare), body mass index and waist and hip circumference were collected at each time point. All outcomes were collected by the same researcher (CTMD) across all time points, following a standardised process.

## Cost-effectiveness

A within-trial cost-utility analysis was conducted to estimate the cost-effectiveness of the intervention in comparison with the control. The outcome measure was quality-adjusted life years (QALYs), which was estimated using self-reported EQ-5D-3L scores, as has been used by previous studies [33–35]. Costs included standard categories of health-service utilisation recommended in UK cardiac rehabilitation economic evaluations, including primary care, outpatient appointments, inpatient care, accident and emergency attendances, and pharmacological treatments. Indirect costs, such as productivity loss or informal care, were not included. The average cost of each arm was determined.

## Statistical analyses

Based on our previous research [24], we anticipated 100 participants, at 80% power, would be required to identify a difference of 2500-step/day (with a standard deviation [SD] of 3500) between the intervention group compared to the control group across the study period, allowing for 15% dropout [24]. This assumed an intra-class correlation co-efficient (ICC) of 0.01, with 15 clusters and 7 individuals in each cluster. However, due to the constraints arising from changes in CR delivery, as explained above, cluster analysis was run at the class level in the end, potentially reducing the study power. Two primary outcomes were defined: daily step counts and time spent in moderate-to-vigorous PA (MVPA) per day. The sample size calculation was based on expected changes in daily step counts, considered the most clinically relevant outcome. No formal adjustment for multiplicity was applied, given the exploratory nature of the study and the limited sample size. All analyses were conducted on participants with valid outcome data. For accelerometer-based outcomes, participants were included if they had at least four valid days of wear time, including at least one weekend day, with a minimum of 10 hours/day, to ensure data were reflective of habitual activity.

ICCs across clusters were calculated in R v4.4.1 using the "psych", "matrix", and "lme4" packages. All other statistical analysis was performed using IBM SPSS Statistics 29.0.2.0. General descriptive statistics such as mean (± SD), median (± interquartile range [IQR]) and number (percentage) were used where appropriate. Comparison of baseline measurements between conditions (t-tests, chi squared, or non-parametric equivalent, as appropriate) confirmed no initial differences. Change in primary outcomes, i.e., PA measures (daily MVPA and daily step counts) were not normally distributed nor transformable; therefore, Mann-Whitney U tests were used to test the effect of condition (control v intervention) on changes in PA from baseline to 6 months later. Additionally, we ran ANOVA models, which included the following factors: condition (with two categories: control and intervention); timepoint (with three categories: baseline, 12 weeks, and 6 months); and their interaction (condition*timepoint), allowing us to test whether these factors predicted daily MVPA or daily step counts. Mean daily MVPA and daily step counts were square root transformed for normality. ANOVAs were conducted using transformed data, whereas the related figure plots original values. ICCs between the CR classes, for key physical and mental health measures, were found to be ubiquitously low (S1 File), indicating that variance between classes was not an important factor. Therefore 'CR Class' was not built into models as a random effect. Data imputation was considered but not conducted due to the limited sample size increasing the risk of Type 1 error.

We calculated descriptive statistics for the QALYs at each timepoint (baseline, 12 weeks, 6 months) by condition (control or intervention). Incremental cost-effectiveness ratios were not calculated because no differences were found in the QALYs between the intervention and control conditions (see results section). The QALYs within the conditions were not normally distributed. Therefore, we compared the QALYs using the Friedman test for repeated measures. Participants with missing values at one or more time points (N = 23) were removed. The statistical analysis plan, including sample size calculations, estimation of intra-class correlation coefficients, and selection of analytic methods, was developed in consultation with an experienced statistician (HMA) to ensure robustness and appropriateness of the analyses. Some participants were lost to follow-up, primarily due to illness, non-return of devices, or disengagement from routine CR. Given the modest sample size, a formal dropout analysis was not conducted to avoid the risk of spurious findings.

### Ethical approval

The trial was registered (ClinicalTrials.gov, Registration date: 26-01-2023, ID: NCT05705310), and was approved by the Institute of Nursing and Health Research Governance Filter Committee (Ulster University: RG3_20-1-3.Z). All participants gave written informed consent.

## Results

### Participant characteristics

Ninety-six participants were recruited for the study between 6 March 2023 and 16 February 2024 (control: n = 44, intervention: n = 52). Participants were 65.04 ± 8.38 (mean ± SD) years old, the majority of whom were male (75%). Full details of recruitment, attrition and data validity are depicted in the CONSORT flowchart (S1 Fig; and in S1 and S2 Tables). Baseline demographics for the entire cohort, and broken down by group, are summarised in Table 2, as are baseline measurements for all primary and secondary outcomes in Table 3. No significant differences between the baseline characteristics were found.

Participants were recruited from six different CR classes, across five local leisure centres, within two neighbouring counties. The attendance rate during the 12-week maintenance stage of CR was the same for participants in both the control and intervention groups (i.e., 67%). Participants adhered with the tracking aspect of the intervention 85% of the time, and the goal setting aspect 55% of the time (averages calculated from all intervention participants across all timepoints). Tracking adherence declined over time and with decreased researcher oversight (S3 Table).

**Table 2. Baseline demographics of study cohort.**

| Participant Characteristics | All participants (n = 96) | Control (n = 44) | Intervention (n = 52) | P value |
|---|---|---|---|---|
| **Age (years)**[a] | 65.04 (± 8.38) | 65.14 (± 7.43) | 64.96 (± 9.19) | 0.92 |
| Missing values | *n = 0* | *n = 0* | *n = 0* | |
| **Gender**[b] | | | | 0.65 |
| Female | 24 (25.00%) | 11 (25.00%) | 13 (25.00%) | |
| Male | 71 (73.96%) | 33 (75.00%) | 38 (73.08%) | |
| Missing values | *1 (1.04%)* | *0 (0.00%)* | *1 (1.92%)* | |
| **Education level**[b] | | | | 0.24 |
| Primary education | 3 (3.13%) | 2 (4.55%) | 1 (1.92%) | |
| Secondary education | 47 (48.96%) | 17 (38.64%) | 30 (57.69%) | |
| University | 41 (42.71%) | 22 (50.00%) | 19 (36.54%) | |
| Another possibility | 4 (4.17%) | 3 (6.82%) | 1 (1.92%) | |
| Missing values | *1 (1.04%)* | *0 (0.00%)* | *1 (1.92%)* | |
| **Civil status**[b] | | | | 0.41 |
| Single | 9 (9.38%) | 6 (13.64%) | 3 (5.77%) | |
| Married/In a civil partnership | 73 (76.04%) | 33 (75.00%) | 40 (76.92%) | |
| Widow/Widower | 3 (3.13%) | 2 (4.55%) | 1 (1.92%) | |
| Divorced/Separated | 10 (10.42%) | 3 (6.82%) | 7 (13.46%) | |
| Missing values | *1 (1.04%)* | *0 (0.00%)* | *1 (1.92%)* | |
| **Living alone**[b] | | | | 0.61 |
| Yes | 18 (18.75%) | 9 (20.45%) | 9 (17.31%) | |
| No | 77 (80.21%) | 35 (79.55%) | 42 (80.77%) | |
| Missing values | *1 (1.04%)* | *0 (0.00%)* | *1 (1.92%)* | |
| **Smoking**[b] | | | | 0.46 |
| Yes | 5 (5.21%) | 4 (9.09%) | 1 (1.92%) | |
| No, but I used to (<1 year ago) | 8 (8.33%) | 3 (6.82%) | 5 (9.62%) | |
| No, but I used to (>1 year ago) | 31 (32.29%) | 13 (29.55%) | 18 (34.62%) | |
| No, never | 51 (53.13%) | 24 (54.55%) | 27 (51.92%) | |
| Missing values | *1 (1.04%)* | *0 (0.00%)* | *1 (1.92%)* | |

For normally distributed continuous data, data are presented as mean (± standard deviation). For categorical variables, data are reported as counts (with percentage). Missing values refer to data in all rows above where missing values are stated.

[a]Denotes t-tests used to examine differences.

[b]Denotes chi squared tests used to examine differences.

Out of three recorded serious adverse events throughout the entire study, (n = 3 occurred in participants in the intervention condition) only one (n = 1) was possibly intervention-related. In this instance, a participant experienced chest pain whilst undertaking independent exercise.

## Primary outcomes

Condition (control vs intervention) did not effect change in PA (daily MVPA and daily step counts) from baseline to 6 months later (S2 Fig and Table 4). The overall models were not significant (Mann Whitney *U*: daily MVPA: W = 1218.00, p = 0.97; daily step counts: W = 1275.00, p = 0.47), demonstrating that the condition did not explain variability in any change in PA.

**Table 3. Baseline primary and secondary outcome measures of study cohort.**

| Outcome Measurement | All participants (n = 96) | Control (n = 44) | Intervention (n = 52) | P value |
|---|---|---|---|---|
| **Primary** | | | | |
| Daily MVPA (mins)[b] | 37.34 [18.08–57.23] | 38.13 [24.61–54.23] | 35.98 [16.22–61.55] | 0.46 |
| Daily step counts[b] | 5980.06 [3355.34–8604.72] | 6408.90 [4016.07–8412.57] | 5632.71 [3311.47–8749.56] | 0.48 |
| Missing values | *n = 10* | *n = 4* | *n = 6* | |
| **Secondary** | | | | |
| BMI (kg/m²) BMI Classification | 29.05 [25.78–32.37] Overweight | 28.76 [26.31–32.17] Overweight | 29.36 [25.10–32.59] Overweight | 0.93 |
| Hip-waist ratio[a] | 0.98 (± 0.08) | 0.99 (± 0.08) | 0.97 (± 0.08) | 0.13 |
| Systolic BP[b] | 115.57 [105.63–128.00] | 112.25 [105.50–127.75] | 118.25 [106.13–129.13] | 0.45 |
| SBP Classification | Normal | Normal | Normal | |
| Diastolic BP[b] | 74.00 [67.75–79.88] | 74.00 [67.38–80.38] | 74.25 [68.00–79.38] | 0.79 |
| DBP Classification | Normal | Normal | Normal | |
| Resting HR[a] | 68.71 (± 10.42) | 66.97 (± 8.47) | 68.00 (± 11.86) | 0.47 |
| HR Classification | Normal | Normal | Normal | |
| Missing values | *n = 0* | *n = 0* | *n = 0* | |
| MSEQ[a] | 16.17 (± 4.58) | 16.98 (± 4.88) | 15.47 (± 4.23) | 0.11 |
| PCS[b] | 49.68 [42.49–53.15] | 50.29 [44.82–53.00] | 47.57 [39.07–58.03] | 0.20 |
| MCS[b] | 52.17 [42.57–58.05] | 51.18 [39.93–58.03] | 53.52 [44.60–58.27] | 0.42 |
| EQ-VAS[b] | 75.00 [65.00–85.00] | 78.50 [70.00–90.00] | 75.00 [60.00–80.00] | 0.08 |
| WEMWBS[b] | 56.00 [49.00–61.00] | 55.25 [49.50–62.50] | 56.00 [48.00–61.00] | 0.83 |
| WEMWBS category | Moderate | Moderate | Moderate | |
| Missing values | *n = 1* | *n = 0* | *n = 1* | |

For normally distributed continuous data, data are presented as mean (± standard deviation). For non-normally distributed continuous variables, data are presented as median [interquartile range]. Missing values refer to data in all rows above where missing values are stated.

[a] Denotes t-tests used to examine differences.

[b] Denotes Mann-Whitney *U* tests used to examine differences.

Abbreviations: Moderate-Vigorous Physical Activity (MVPA), Body Mass Index (BMI), Blood Pressure (BP), Heart Rate (HR), Marcus Self-Efficacy Questionnaire (MSEQ), Physical Component Score (PCS), Mental Component Score (MCS), EuroQol Visual Analogue Scale (EQ-VAS), Warwick Edinburgh Mental Wellbeing Scale (WEMWBS).

A marginal increase in mean daily MVPA levels and steps were seen for those in the intervention condition from baseline to 12 weeks, with a subsequent drop between the 12 week and 6 month timepoint, while a contrasting pattern was seen for those in the control condition (S4 Fig and S4 Table). Importantly, despite these patterns, a model testing condition, timepoint, and their interaction (condition*timepoint) measure of PA showed no significant difference (ANOVA: daily MVPA: $F_{5,1} = 0.57$, $p = 0.72$; daily steps: $F_{5,1} = 0.25$, $p = 0.94$; S3 Fig).

## Secondary outcomes

Changes in PA self-efficacy from baseline to 6 months did not differ between conditions, nor did any physical or mental health or wellbeing measures (see Table 4).

The median, and IQR for the QALYs reported by each group are presented in S5 Table. S1 Fig present a visualisation of the change in QALYs over time. Median QALYs remained consistent (0.88) in both conditions at all timepoints. Therefore, QALYs did not differ between timepoint for either condition (Friedman tests: control: $X^2_2 = 1.06$, $p = 0.59$; intervention: $X^2_2 = 2.53$, $p = 0.28$). Nevertheless, average cost of health service use per participant during their 6 month study period was

**Table 4. Change in outcome measures of study cohort from baseline to 6 months.**

| Outcome Measurement | Control (n = 44) | Intervention (n = 52) | P value |
|---|---|---|---|
| **Δ Primary** | | | |
| Daily MVPA (mins)[b] | −2.84 [−14.62–12.23] | −2.68 [−11.32–4.60] | 0.97 |
| Daily step counts[b] | −431.83 [−1286.25–1595.57] | −133.79 [−1175.04–929.75] | 0.47 |
| Missing values | n = 16 | n = 15 | |
| **Δ Secondary** | | | |
| MSEQ[a] | 0.58 (± 3.14) | 1.47 (± 3.61) | 0.16 |
| BMI[a] | 0.17 (± 1.04) | 0.06 (± 0.94) | 0.62 |
| Hip-waist ratio[b] | 0.00[−0.45–0.72] | −0.01 [−0.02–0.01] | 0.94 |
| Systolic BP[a] | 5.24 (± 12.02) | 7.65 (± 13.03) | 0.40 |
| Diastolic BP[a] | 1.26 (± 8.53) | 1.55 (± 9.19) | 0.89 |
| Resting HR[b] | −4.50 [−8.50–0.75] | −4.75 [−9.63 − −0.13] | 0.63 |
| PCS[a] | 1.15 (± 6.54) | −0.08 (± 6.49) | 0.99 |
| MCS[a] | 3.44 (± 7.22) | 2.35 (± 6.77) | 0.18 |
| EQ-VAS[b] | 9.50 [0.00–25.00] | 10.00 [0.00–23.75] | 0.86 |
| Missing values | n = 8 | n = 10 | |
| WEMWBS[b] | 4.00 [0.25–8.75] | 2.00 [−1.00–11.00] | 0.28 |
| Missing values | n = 9 | n = 10 | |

For normally distributed continuous data, data are presented as mean (± standard deviation). For non-normally distributed continuous variables, data are presented as median [interquartile range]. Missing values refer to data in all rows above where missing values are stated.

[a]Denotes t-tests used to examine differences.

[b]Denotes Mann-Whitney U tests used to examine differences.

Abbreviations: Change in (Δ), Moderate-Vigorous Physical Activity (MVPA), Body Mass Index (BMI), Blood Pressure (BP), Heart Rate (HR), Marcus Self-Efficacy Questionnaire (MSEQ), Physical Component Score (PCS), Mental Component Score (MCS), EuroQol Visual Analogue Scale (EQ-VAS), Warwick Edinburgh Mental Wellbeing Scale (WEMWBS).

£444.97 for individuals in the intervention condition, whereas for individuals in the control condition that value was more than double, at £1,015.34. The mean cost of the intervention per participant was £74.68. Further cost-effectiveness analyses were not conducted because there was no significant difference in QALYs between the two conditions.

## Discussion

This RCT explored the effects of a behaviour change intervention on CHD patients, with outcomes assessed 6 months after they began attending maintenance CR. The findings indicate that the intervention did not lead to an increase in PA, and PA levels of those receiving the intervention did not differ significantly from those receiving standard care. The lack of difference between conditions held true at all timepoints. Furthermore, changes in participants self-rated PA self-efficacy, quality of life, physical and mental health, and mental wellbeing did not differ between conditions. These findings suggest that adding a behaviour change intervention to current maintenance stage CR did not impact any quantitative measures of PA or related health outcomes. The study was guided by a steering group comprising clinical experts, researchers, key stakeholders and patient representatives, who contributed to all aspects of study design, intervention development, and implementation, ensuring that the intervention was relevant, feasible and acceptable to participants.

In terms of offering explanation for the findings, the observed baseline PA levels of participants in our study were much higher than expected, and than those in previous studies [36,37]. However, they are in line with a recent randomised controlled trial that also found high baseline PA levels in patients with coronary artery disease [38]. In fact, participants mean/median activity levels at baseline, in both the control and intervention conditions, were already surpassing the BACPR and

WHO guidelines for MVPA. Therefore, the lack of any significant increase in PA levels during the course of the intervention may be explained by the limited realistic potential or need for further improvement. Additionally, the interval between completion of core CR and enrolment in the study varied considerably due to COVID-19–related service pauses and delays, ranging from approximately 3 months to up to 2 years post–cardiac event. This variability may have influenced participants' readiness for further behaviour change. The findings from a nested process evaluation will further explore participants' experiences of the intervention and their reasons for a lack of change in PA. This will be published separately.

Previous research has shown that maintenance of a healthy behaviour, and PA in particular, often declines without ongoing support following CR [18,38,39]. Typically, the CR is available for a 12 week period. However, due to undersubscription of the CR classes in most centres during the period of the study, individuals who wished to continue to attend CR were offered the opportunity to do so beyond 12 weeks. This additional support, through extended access, may explain why the expected reduction in PA in the control condition was not observed, as has been seen in other studies with longer timeframes and follow-up prompts [40]. Consequently, the potential benefits of the behaviour change intervention may have been masked by the prolonged access to CR in both groups.

It is important to highlight that no significant decrease in PA was observed throughout the duration of the current study, including at the last outcome assessment, which typically aligned closely with one year after an individual experienced their cardiac event. Therefore, although the independent impact of the behaviour change intervention remains unclear, these findings highlight potential mechanisms to sustain PA levels, through extended support in maintenance CR programmes.

While this study suggests that continued access to maintenance CR can help individuals achieve and sustain guideline-level PA, such access is not a practical or scalable solution for all CR patients. BACPR competencies identify forward planning strategies to support individuals in establishing an exit strategy from CR [7], yet, building confidence and autonomy to exercise independently is an area that needs to be further developed in practice. Therefore, greater emphasis should be placed on empowering individuals through behaviour change interventions that build self-efficacy and independence [41,42]. Additionally, it is possible that the individuals who are more likely to enrol in, and remain in, a study aimed at monitoring and self-motivating PA, are more likely to have high levels of activity already [43], a theory supported by our high mean/median baseline levels of PA. Further consideration should be given to developing interventions that target and appeal specifically to individuals at the maintenance stage of their CR who have lower initial activity levels and are not meeting BACPR guidelines. Target individuals could be identified by cardiac nurses during the referral process from core CR or through the addition of a screener, such as the general practice PA questionnaire, at the beginning of maintenance CR. In addition, efforts to determine perceived barriers to participating in maintenance CR, for those who decline, may be of value.

## Study limitations

Although originally designed as a stepped-wedge trial, the implemented design functioned as a cluster randomised trial with a stepped implementation, which may limit comparability to other stepped-wedge studies and could affect generalisability of findings. Due to the nature of the study, it was not possible to blind participants to which condition they had been assigned, which may influence subsequent behaviour. Additionally, the planned number of clusters was not achieved, reducing the statistical power of the study and limiting the certainty of the findings. Adherence to the intervention waned over time, which may diminish its intended effect. However, these adherence rates are representative of true behaviours, therefore, provide valuable insight into what outcomes can realistically be expected. Variation in adherence to specific behaviour change components, particularly early tasks such as goal-setting and self-monitoring, likely diluted the observed intervention effect and should be considered when interpreting the findings.

Given the pragmatic, service-embedded nature of the study, there is potential for selection bias, as participants who chose to enrol and remain in the study may have been healthier, more motivated, or more physically active than the

general CR population. While consecutive recruitment and baseline comparisons suggest groups were broadly comparable, this limitation should be considered when generalising the findings. In addition, two primary PA outcomes (daily MVPA and daily step counts) were assessed; no formal adjustment for multiplicity was applied due to the exploratory nature of the study and limited sample size, which may increase the risk of Type I error.

There were adjustments to service provision from the time the study was designed to the time the study was conducted. Rolling recruitment varied from the original protocol and caused necessary deviation from the planned provision of discussion groups; resulting in fewer discussions per participant, as well as smaller groups. Potential contamination cannot be fully excluded as some classes were delivered at the same site; however, small group and individual delivery of the intervention likely minimized cross-condition influence. Additionally, uncapped access to CR class beyond 12 weeks in most classes varied from service provision at time of study design. Therefore, the majority of participants in both control and intervention conditions had continued support that we had not anticipated. While continued access is helpful for participants, it likely caused considerable deviation from the expected PA levels, particularly for our control group. Moreover, although all participants continued attending standard maintenance CR sessions alongside the behaviour change intervention, attendance was highly variable. These differences in ongoing participation, while not included as a covariate, may have influenced individual behaviour change trajectories and the observed effects of the intervention. Finally, CR programme delivery in reality was 1 day per week, inconsistent with evidence-based guidelines of 2 days per week.

## Conclusions

A behaviour change intervention added to a usual maintenance stage CR programme, did not lead to changes in PA, or any measures of health or wellbeing measured in this study. However, high PA levels were maintained throughout the duration of the study by both control and intervention groups. These findings should be interpreted in the context of study limitations, including the reduced number of clusters, potential selection bias, variability in CR participation, and deviations from the originally planned stepped-wedge design. Follow-up studies should aim to disentangle the effects of the behaviour change intervention, and the benefits of continued access to maintenance CR classes.

## Supporting information

**S1 Table. Intra-class correlation co-efficients (ICCs) for key baseline physical and mental health measurements.**
(DOCX)

**S2 Table. Recruitment, data completeness and attrition rates across all timepoints.**
(DOCX)

**S3 Table. Adherence rates of physical activity (PA) tracking across the course of the study.**
(DOCX)

**S4 Table. Mean (± standard deviation) for daily moderate-vigorous physical activity (MVPA) and step counts at each timepoint.**
(DOCX)

**S5 Table. Median [and interquartile range] for Quality Adjusted Life Years at each timepoint.**
(DOCX)

**S1 Fig. CONSORT participant flowchart of the STRENGTH cluster randomised controlled trial.**
(DOCX)

**S2 Fig. Change in daily physical activity for control versus intervention conditions from baseline to 6 months later.**
(DOCX)

**S3 Fig. Mean daily physical activity for each condition, across all timepoints.**
(DOCX)

**S4 Fig. Boxplots represent median and IQR of QALYs at each timepoint.**
(DOCX)

**S1 File. CONSORT checklist.**
(PDF)

**S2 File. TIDieR checklist.**
(DOCX)

## Acknowledgments

We thank the members of our steering committee who are not also named authors (NB, GC, AS, RJ) for guidance and feedback throughout the course of the study. We are also very grateful to our host exercise facilities and CR instructors for facilitating the research, and lastly to the CR participants themselves for sharing their time, effort and insight. HMcA is funded by the HSC Research and Development Division of the Public Health Agency, through the Northern Ireland Public Health Research Network.

## Author contributions

**Conceptualization:** Mark A. Tully, Nicole E. Blackburn.

**Formal analysis:** Clare T.M. Doherty, Mark A. Tully, Jason J. Wilson, Leonie Heron, Helen McAneney, Nicole E. Blackburn.

**Funding acquisition:** Mark A. Tully, Jason J. Wilson, Nicole E. Blackburn.

**Investigation:** Clare T.M. Doherty, Nicole E. Blackburn.

**Methodology:** Clare T.M. Doherty, Mark A. Tully, Jason J. Wilson, Nicole E. Blackburn.

**Writing – original draft:** Clare T.M. Doherty, Mark A. Tully, Jason J. Wilson, Helen McAneney, Nicole E. Blackburn.

**Writing – review & editing:** Clare T.M. Doherty, Mark A. Tully, Jason J. Wilson, Leonie Heron, Helen McAneney, Victoria Irving, Lisa Spratt, Rachel O'Reilly, Kim Kensitt, Nicole E. Blackburn.

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
