## [Decision Letter · Decision Letter 0]

15 Sep 2025

Dear Dr. Blackburn,

We look forward to receiving your revised manuscript.

Kind regards,

Sascha Köpke

Academic Editor

PLOS ONE

“NEB

Heart Research UK

GR2686

https://heartresearch.org.uk/

No”

“We thank Heart Research UK for funding this research. Many thanks also go to the members of our steering committee who are not also named authors (NB, GC, AS, RJ) for guidance and feedback throughout the course of the study. We are also very grateful to our host exercise facilities and CR instructors for facilitating the research, and lastly to the CR participants themselves for sharing their time, effort and insight. HMcA is funded by the HSC Research and Development Division of the Public Health Agency, through the Northern Ireland Public Health Research Network.”

“NEB

Heart Research UK

GR2686

https://heartresearch.org.uk/

No”

5. We notice that your supplementary figures and tables are included in the manuscript file. Please remove them and upload them with the file type 'Supporting Information'. Please ensure that each Supporting Information file has a legend listed in the manuscript after the references list.

Reviewers' comments:

Reviewer's Responses to Questions

**Comments to the Author**

1. Is the manuscript technically sound, and do the data support the conclusions?

Reviewer #1: No

Reviewer #2: Partly

Reviewer #3: Partly

2. Has the statistical analysis been performed appropriately and rigorously?

Reviewer #1: I Don't Know

Reviewer #2: I Don't Know

Reviewer #3: Yes

3. Have the authors made all data underlying the findings in their manuscript fully available?

Reviewer #1: Yes

Reviewer #2: Yes

Reviewer #3: Yes

4. Is the manuscript presented in an intelligible fashion and written in standard English?

Reviewer #1: Yes

Reviewer #2: Yes

Reviewer #3: Yes

Reviewer #1: Thank you for giving me the oppertunity to review this paper.

Abstract

• It is misleading to state that 96 participants were randomised, as randomisation was conducted at the class (cluster) level. A more correct description would be that six cardiac rehabilitation (CR) classes were randomised, with participants following the allocation of their class.

• The study is therefore better classified as a cluster randomised trial rather than a true stepped-wedge trial, particularly since the stepped-wedge element is not actually utilised in the analyses.

Background

• Much of the cited literature is outdated (e.g. MacKay & Mensah 2004). The organisation and delivery of cardiac rehabilitation has changed substantially since then.

• Recent systematic reviews, especially the Cochrane review by Dibben 2021, as well as newer RCTs, should be included. References such as Dalal 2015 are insufficient as the main evidence base.

• The background section is generally weak and should be rewritten with up-to-date literature.

Methods

• The authors cite Hemming 2015, which is relevant, but more recent methodological references should be used (e.g. Chan, Leyrat & Eldridge 2024, Trials; Fan Li et al. 2022, World Neurosurgery).

• It is unclear whether all CR classes took place in different leisure centres or in the same centre. If the same centre was used, there is a considerable risk of contamination bias, as health professionals in control classes may have known the intervention.

• Table 1: Recruitment period 3 is not used and should be deleted. More generally, the rationale for choosing a stepped-wedge design is not clear, given that its features are not applied in the analyses.

Statistical analyses

• The sample size calculation is insufficiently described and cannot be reproduced. It is unclear how the ICC was estimated. A statistician should review this section.

• The study initially planned for 15 clusters, but only 6 CR classes were included (with 4 implemented as planned). This is a major limitation, as the required sample size was not achieved, severely reducing power and validity.

• See Chan, Leyrat & Eldridge (2024) for proper handling of cluster stepped-wedge sample size and analyses.

Results

• The results section should start by clearly stating the number of classes and participants in each class. There is inconsistency between the text (line 229: six classes) and Table 1 (4+2). This needs clarification.

• Participants appear healthier, better educated, and more physically active than the typical CR population. This selection bias substantially limits generalisability.

• A dropout analysis is missing: baseline values for those not included or who dropped out should be presented (age, sex, education, living situation, comorbidity, etc.).

• It is also unclear who received further CR after the 12 weeks (lines 300–306). At minimum, numbers and extent of CR participation should be reported. Without these data, it is impossible to evaluate the independent effect of the intervention.

Discussion & Limitations

• The discussion mentions high baseline activity levels and extended CR access as possible explanations, but these should have been addressed as design weaknesses already in the methods.

• The most important limitations are:

1. Misclassification as a stepped-wedge trial.

2. Insufficient statistical power due to too few clusters.

3. Risk of contamination if intervention and control classes were delivered at the same site.

4. Strong selection bias (participants not representative of the CR population).

5. Missing dropout analysis.

6. Lack of clarity on extended CR after the intervention.

7. Outdated background literature.

Conclusion

The study suffers from serious methodological flaws: it is underpowered, the participants are not representative, the design is misclassified, and the analyses do not utilise stepped-wedge methodology. In addition, the background is outdated, and essential analyses such as dropout characteristics and extended CR participation are missing.

For these reasons, I find the manuscript unsuitable for publication in its current form.

Reviewer #2: Thank you for the opportunity to review this manuscript, describing the results of a stepped-wedge RCT of a behaviour change intervention added to maintenance cardiac rehabilitation compared to standard maintenance CR alone. The manuscript is written well, and although no effects were observed, the findings remain important. However, several key details about the intervention and control conditions are missing, which impacts the reliability and replicability of the study. The authors may wish to consider the following points:

General:

• Consistency of terminology. Throughout the manuscript, the intervention is described in a variety of ways, (e.g. “behaviour change intervention” in the title, “self-management intervention” in the abstract, and “multi-component intervention” in the background. A consistent term should be used throughout.

Background:

• It could be made clearer that cardiac rehabilitation aims to increase both physical fitness and physical activity levels through its various components.

• A more detailed discussion of why an increase in PA is harder to achieve/sustain would strengthen the rationale. This could be linked to the potential value of interventions that incorporate behaviour change techniques.

Methods:

• Please provide more detail of the prior CR (phase III) received by participants? For example, how long was the gap (if any) between completion of CR and the start of the maintenance intervention?

• Please provide more details of what the standard CR maintenance programme involved. How many sessions per participants expected to attend per week? How long did the sessions last? Were they delivered in-person? Did the sessions involve participation in an exercise programme? If so, what was the exercise prescription?

• As the intervention is described as a behaviour change intervention, I would expect to see some description of the behaviour change techniques used, and the mechanisms by which they were expected to work.

• The duration and content of the behaviour change intervention is not clear. What took place for the first six weeks before the pedometers were handed out? How many group discussion sessions were there? How long were these sessions? What was the typical number of participants involved in each class?

• Was there any patient and public involvement in the intervention development or other aspects of the study?

Results/Discussion:

• Did the participants who dropped out of the study differ from those who remained?

• If the information is available, it would be useful to know what CHD diagnoses the participants had.

• Please specify what costs were included in the calculations for health service use?

• Was any adjustment made for multiplicity, given that there were two primary outcomes? If not, this should be acknowledged in the limitations.

• How much might the varying levels of adherence to the intervention have impacted the results? E.g., 55% goal setting.

Reviewer #3: This is evaluating the effectiveness of a multi-component intervention incorporating current evidence from previous research and applying it to a maintenance stage CR programme to promote sustained changes in PA following completion.

Some comments for the authors consideration.

The sample size is based on one of the outcomes - yet the primary outcome section states they are potentiall two, i.e steps per day and time spent in moderate vigorous PA (MVPA) per day. Either the sample size needs to reflect the both outcomes, or clarity whether outcome number 2 is a secondary outcome?

Under study design - suggest to start the sentence with the stating what the study design, e.g "This is a stepped-wedged cluster randomised trial". details regarding the reporting guidelines - this should actually be under statistical analyses as you are stating how the study will be report.

In terms of recruitment - line 59, there is mention of 6 CR classes across 2 counties, with table 1 showing the allocation sequence, across these counties, can you have more than one CR clas within each county? Can the distinct randomisation unit be made clear? Ah I see its mentioned in line 229 - this should probably be in the methods section.

Under statistical analysis - it would be good to define population for analyses, since line 143 suggest, a criteria needed to be met.

Can an explanation be given to explain the delay in recruitment, i.e line 60 says planned recruitment period was Mar 2022, but actual date reported in results section says Mar 2023 (line 203).

**Do you want your identity to be public for this peer review?** For information about this choice, including consent withdrawal, please see our Privacy Policy

Reviewer #1: No

Reviewer #2: No

Reviewer #3: No

---

## [Author Response · Author response to Decision Letter 1]

15 Jan 2026

We would like to thank all reviewers for their thorough and constructive feedback. We have carefully revised the manuscript in response to each point raised. I have included our full rebuttal within the cover letter where we provide a detailed response to all comments and indicate where changes have been made within the manuscript, which are clearly identified within the version with tracked changes.

We sincerely thank all reviewers for their insightful and constructive feedback. In response, the manuscript has undergone substantial revision to:

- Address methodological concerns, including sample size, ICC, and cluster structure;

- Clarify the study design as a cluster randomised trial with a stepped implementation sequence, and explain why a formal stepped-wedge analytic approach was not applied;

- Enhance transparency regarding recruitment, class delivery, prior CR participation, intervention content, and potential sources of bias;

- Strengthen reporting of outcomes, dropout analyses, adherence, and cost-effectiveness methods; and

- Explicitly acknowledge study limitations, including variability in participant adherence, potential contamination, and reduced statistical power due to the limited number of clusters.

These revisions ensure that the manuscript more accurately reflects the study’s real-world context, scope, and constraints, while providing a clear and transparent account of the intervention’s implementation and findings.

Warmest regards,

Dr Nicole E. Blackburn

---

## [Decision Letter · Decision Letter 1]

5 Mar 2026

The STRENGTH Study: A cluster randomised controlled trial of the effect of a behaviour change intervention added to cardiac rehabilitation on physical activity adherence

PONE-D-25-39199R1

Dear Dr. Blackburn,

We’re pleased to inform you that your manuscript has been judged scientifically suitable for publication and will be formally accepted for publication once it meets all outstanding technical requirements.

Kind regards,

Sascha Köpke

Academic Editor

PLOS One

Additional Editor Comments (optional):

Reviewers' comments:

Reviewer's Responses to Questions

**Comments to the Author**

Reviewer #1: All comments have been addressed

Reviewer #3: All comments have been addressed

2. Is the manuscript technically sound, and do the data support the conclusions?

Reviewer #1: Partly

Reviewer #3: Yes

3. Has the statistical analysis been performed appropriately and rigorously?

Reviewer #1: Yes

Reviewer #3: Yes

4. Have the authors made all data underlying the findings in their manuscript fully available?

Reviewer #1: Yes

Reviewer #3: Yes

5. Is the manuscript presented in an intelligible fashion and written in standard English?

Reviewer #1: Yes

Reviewer #3: Yes

Reviewer #1: The authors have addressed my questions from the first review round. On re-reading the manuscript, I note that self-efficacy does not change significantly between the groups. It is therefore possible that the issue lies with the specific intervention rather than with the underlying hypothesis. I leave it to the editor to decide whether this point should be discussed, as I did not raise it in the first review.

Reviewer #3: (No Response)

**Do you want your identity to be public for this peer review?** For information about this choice, including consent withdrawal, please see our Privacy Policy

Reviewer #1: No

Reviewer #3: No

---

## [Editor Report · Acceptance letter]

PONE-D-25-39199R1

PLOS One

Dear Dr. Blackburn,

I'm pleased to inform you that your manuscript has been deemed suitable for publication in PLOS One. Congratulations! Your manuscript is now being handed over to our production team.

Kind regards,

on behalf of

Professor Sascha Köpke

Academic Editor

PLOS One